# Lattices-Inspired CP-ABE from LWE Scheme for Data Access and Sharing Based on Blockchain

Taowei Chen [1,2], Zhixin Ren [1,3,*], Yimin Yu [1,3], Jie Zhu [4] and Jinyi Zhao [5]

1   School of Information, Yunnan University of Finance and Economics, Kunming 650221, China; twchen@ynufe.edu.cn (T.C.); yym@ynufe.edu.cn (Y.Y.)
2   Yunnan Key Laboratory of Blockchain Application Technology, Kunming 650233, China
3   Institute of Intelligent Application, Yunnan University of Finance and Economics, Kunming 650221, China
4   Division of Science & Technology Administration, Yunnan University of Finance and Economics, Kunming 650221, China; zz0975@ynufe.edu.cn
5   School of Logistics and Management Engineering, Yunnan University of Finance and Economics, Kunming 650221, China; zhaojinyi@stu.ynufe.edu.cn
*   Correspondence: 202202110688@stu.ynufe.edu.cn

**Abstract:** To address the quantum attacks on number theory-based ciphertext policy attribute-based encryption (CP-ABE), and to avoid private key leakage problems by relying on a trustworthy central authority, we propose a lattice-inspired CP-ABE scheme for data access and sharing based on blockchain in this paper. Firstly, a CP-ABE-based algorithm using learning with errors (LWE) assumption is constructed, which is selective security under linear independence restriction in the random oracle model. Secondly, the blockchain nodes can act as a distributed key management server to offer control over master keys used to generate private keys for different data users that reflect their attributes through launching transactions on the blockchain system. Finally, we develop smart contracts for proving the correctness of proxy re-encryption (PRE) and provide auditability for the whole data-sharing process. Compared with the traditional CP-ABE algorithm, the post-quantum CP-ABE algorithm can significantly improve the computation speed according to the result of the functional and experimental analysis. Moreover, the proposed blockchain-based CP-ABE scheme provides not only multi-cryptography collaboration to enhance the security of data access and sharing but also reduces average transaction response time and throughput.

**Keywords:** post-quantum cryptography; blockchain; ciphertext policy attribute-based encryption; data sharing; privacy protection





## 1. Introduction

With the increasing popularity of blockchain technology, blockchain has been widely used in data access and sharing in weak-trusted or no-trusted networks. However, privacy data protection on the blockchain has become a challenge due to the characteristics of "non-tampering" and "open and transparent". In recent years, many scholars have put forward some new solutions for privacy protection and data sharing on the blockchain [1–6], among which attribute-based encryption algorithm has been widely used in various schemes on the blockchain, such as in data traceability [1], cloud storage [7], medical data sharing [8,9], power systems [10] and Internet of Things [11] due to its advantages of "one-to-many" encryption, fine-grained access control and so on.

The attribute-based encryption (ABE) originated from the fuzzy identity-based encryption [12] proposed by Sahai and Waters in 2005 and then developed into ABE. In key policy attribute-based encryption [13] (KP-ABE), the ciphertext is associated with the attribute and the key with the access policy. However, in ciphertext policy attribute-based encryption (CP-ABE), the key is associated with the attribute and the ciphertext with the access policy. The CP-ABE allows the data owner to freely formulate the access control

policy and is more suitable for the distributed storage environment and uncertain deciphering [14]. In recent years, the main research on the ABE mainly focused on computation efficiency [15,16], access policy and attribute hiding [17], and identity management [18,19]. In 2007, Bethencourt et al. [20] proposed a system for the CP-ABE algorithm which allows policies to be expressed as any monotonic tree access structure and is resistant to collusion attacks in which an attacker might obtain multiple private keys. In addition, the "one-to-many" encryption and fine-grained access control could be achieved this way. However, it is proved secure under the generic group heuristic. In 2011, Waters [21] proved the security of the CP-ABE under the standard model and put forward a CP-ABE adopting the linear secret sharing scheme, which improved the efficiency significantly. In 2012, Okamoto [22] et al. proposed the first unbounded inner product ABE scheme, which lifted the restrictions on the predicate terms and size attributes of the previous ABE scheme. In 2013, Gorbunov [23] proposed an ABE scheme based on the polynomial logic circuit. This scheme was designed to resist collusion attacks effectively, and its public parameter and ciphertext size increased linearly with the circuit depth. This scheme made it possible to transform from Boolean formula-based to circuit-based, which had better security by nature. In 2014, Waters [24] proposed an Online-Offline ABE scheme to address the computational bottleneck of encryption and key generation. This scheme was inspired by the ABE scheme put forward by Rouselakis et al. [25], and developed new techniques for ABE that split the computation for encryption into two phases, thus reducing the computation consumption in the online stage.

Recently, the blockchain-enhanced CP-ABE scheme using elliptic curve cryptography (ECC) has become a popular research area due to its potential to provide secure data access and sharing without involving a third party. In 2020, Yuwen Pu [26] presented a privacy-preserving, recoverable, and revocable edge data-sharing scheme based on blockchain technology to achieve attribute revocation in ciphertext-policy attribute-based encryption (CP-ABE), which can protect user's privacy and resist many attacks. Sheng Gao [27] proposed a new trustworthy, secure ciphertext policy and attribute hiding access control scheme based on blockchain to achieve trustworthy access while guaranteeing the privacy of policy and attribute. Meanwhile, Xuanmei Qin [28] proposed a scheme that manages each attribute across different domains, eliminates the single-point bottleneck of the existing multi-authority blockchain-based CP-ABE schemes, and reduces computation and communication overhead between the user and the multiple attribute authorities. In 2022, Guofeng Zhang [29] proposed a secure and trusted agricultural product traceability system (BCST-APTS) supported by blockchain and CP-ABE encryption technology, thereby ensuring the efficient sharing and supervision of data stored in the Permissioned Blockchain.

However, most of the aforementioned works based their security on cryptographic assumptions related to bilinear maps. It is very natural to seek for solutions to the known attacks on group-based constructions by quantum computers. To address the threat of quantum computing, lattice-based cryptography is a promising approach that provides a new set of assumptions based on finding short vectors in lattices and is believed to be hard for quantum computers. Additionally, new cryptographic protocols have been proposed which rely on the hardness of solving certain lattice problems, including Learning with Errors (LWE). These protocols are expected to provide strong protection against the attacks of quantum computers. At present, lattice-difficulty problems that are provably secure mainly consist of small integer solution problems (SIS) and learning with errors problems (LWE). These two difficult problems are as hard as approximate lattice problems from the worst case to the average case reduction. Recently, several lattice-based encryption schemes have been proposed consecutively, which mainly focus on identity-based encryption [30–32] digital signature [33], and zero-knowledge proof [34]. In May 2021, Datta et al. [35] constructed an ABE algorithm based on ciphertext policy according to the LWE-difficulty problem and realized a CP-ABE scheme that could resist quantum attacks.

In addition, for a comprehensive analysis of our security measures, we considered the potential impacts of active and passive side-channel attacks (SCA), fault attacks, and power

analysis attacks [36]. We recognize the increasing importance of Post-Quantum Cryptography (PQC) in secure applications, such as smartphones and blockchain systems, as it replaces traditional ECC/RSA algorithms. Therefore, we studied relevant literature [37–39]. Furthermore, we acknowledged the significance of NIST lightweight standardization and considered fault attacks as a consideration for side-channel attacks, referring to the research work of Mozaffari-Kermani et al. [40–42] in this field.

The contributions of this scheme are as follows:

(a) Form the LWE-CPABE algorithm suitable for blockchain to resist the quantum attacks, improve the CP-ABE scheme proposed by Datta, design the CP-ABE algorithm supporting policy update, and realize the dynamic access control of the data;

(b) Use a threshold proxy re-encryption scheme following a key encapsulation mechanism (KEM) approach to implement distributed key management for the LWE-CPABE algorithm, design the transaction generation algorithm and transaction verification contract, and realize the correctness and integrity verification of the intra-transaction data and the outsourcing storage data;

(c) Carry out the security analysis and simulation experiment, and the results show that this scheme is secure and efficient and is suitable for distributed data sharing and access control.

## 2. Preliminaries

### 2.1. Notations

In this paper, we use the following notations and conventions. The security parameter is denoted by $\lambda$. A function $\textit{negl} : \mathbb{N} \to \mathbb{R}$ is considered negligible if it decreases faster than any inverse-polynomial function. Formally, for every constant $c > 0$, there exists an integer $N_c$ such that $\textit{negl}(\lambda) \leq \lambda^{-c}$ for all $\lambda > N_c$, where $[n] = \{1, \cdots, n\}$ is the negligible function.

The abbreviation PPT represents probabilistic polynomial-time algorithms. When sampling from a distribution $\mathcal{X}$, we use $x \leftarrow \mathcal{X}$ to denote the random value sampled from the $\mathcal{X}$ distribution. For a set $X$, the notation $x \leftarrow X$ indicates that $x$ is sampled uniformly from the elements of set $X$. Bold lowercase letters, such as $\boldsymbol{v}$, represent vectors, while bold uppercase letters, such as $\boldsymbol{M}$, represent matrices. By default, all vectors in this paper are assumed to be row vectors. In the context of matrices, the $j$-th row is denoted as $\boldsymbol{M}_j$, and $\boldsymbol{M}_J$ represents the submatrix of $\boldsymbol{M}$ composed of all rows indexed by $j \in J$, where $J$ is a set of row indexes. For a vector $\boldsymbol{v}$, we use $\|\boldsymbol{v}\|$ to denote its $\ell_2$-norm, and $\|\boldsymbol{v}\|_\infty$ represents the $\ell_\infty$-norm of the vector.

For an integer $q \geq 2$, we let $\mathbb{Z}_q$ denote the ring of integers modulo $q$. We use $\mathbb{Z}_q$ to represent integers in the range $(-q/2, q/2]$.

### 2.2. B-Bounded

For a family of distributions $\mathcal{D} = \{\mathcal{D}_\lambda\}_{\lambda \in \mathbb{N}}$ over the integers and there is a boundary $B = B(\lambda) > 0$. If we denote that $\mathcal{D}$ is $B$-bounded for every $\lambda \in \mathbb{N}$, it holds that:

$$\Pr_{x \leftarrow \mathcal{D}_\lambda}[|x| \leq B(\lambda)] = 1 \tag{1}$$

**Lemma 1.** *Let $B_1 = B_1(\lambda)$ and $B_2 = B_2(\lambda)$ be positive, and let $\mathcal{D} = \{\mathcal{D}_\lambda\}_\lambda$ be a B-bounded distribution family of $B_1$. We let $\mathcal{U} = \{\mathcal{U}_\lambda\}_\lambda$ denote the uniform distribution over $[-B_2(\lambda), B_2(\lambda)]$. If there is a negligible function $\textit{negl}(\cdot)$ such that for all $\lambda \in \mathbb{N}$ it holds that:*

$$B_1(\lambda)/B_2(\lambda) \leq \textit{negl}(\lambda) \tag{2}$$

*and then the distribution family $\mathcal{D} + \mathcal{U}$ and $\mathcal{U}$ are statistically indistinguishable.*

*2.3. Leftover Hash Lemma*

**Lemma 2.** *(leftover hash lemma) Let $n : \mathbb{N} \to \mathbb{N}$, $q : \mathbb{N} \to \mathbb{N}$, $m > (n+1)\log q + \omega(\log n)$, and we donte $k = k(n)$ as some polynomial. Then, the following two distributions are statistically indistinguishable:*

$$\begin{aligned}
\mathcal{D}_1 &\equiv \left\{ (A, AR) \Big| A \leftarrow \mathbb{Z}_q^{n \times m}, R \leftarrow \{-1, 1\}^{m \times k} \right\}, \\
\mathcal{D}_2 &\equiv \left\{ (A, S) \Big| A \leftarrow \mathbb{Z}_q^{n \times m}, S \leftarrow \mathbb{Z}_q^{n \times k} \right\}.
\end{aligned} \tag{3}$$

*2.4. Lattice*

Here, we briefly describe the necessary background on lattices. A lattice $\mathcal{L}$ is defined as the discrete addition subgroup with the dimension being $m$ in $\mathbb{R}^m$, assuming the positive integers $n$, $m$, $q$ and the matrix $A \in \mathbb{Z}_q^{n \times m}$, and let $\lambda_q^\perp(A)$ represent the lattice $\left\{ x \in \mathbb{Z}^m \Big| Ax^\perp = 0^\perp \bmod q \right\}$. For $u \in \mathbb{Z}_q^n$, we let $\lambda_q^u(A)$ denote as the coset $\left\{ x \in \mathbb{Z}^m \Big| Ax^\perp = u^\perp \bmod q \right\}$.

2.4.1. Discrete Gaussians

Let $\sigma$ be any positive real number. Generating a Gaussian distribution $\mathcal{D}_\sigma$ is defined by the probability distribution function (PDF) $\rho_\sigma(x) = \exp\left(-\pi \|x\|^2 / \sigma^2\right)$. For any discrete set $\mathcal{L} \in \mathbb{R}^m$, we define $\rho_\sigma(\mathcal{L}) = \sum_{x \in \mathcal{L}} \rho_\sigma(x)$. The discrete Gaussian distribution $\mathcal{D}_{\mathcal{L}, \sigma}$ over the lattice $\mathcal{L}$ with the parameter $\sigma$ is defined by PDF $\rho_{\mathcal{L}, \sigma}(x)$:

$$\rho_{\mathcal{L}, \sigma}(x) = \rho_\sigma(x) / \rho_\sigma(\mathcal{L}) \tag{4}$$

**Lemma 3.** *If the parameter $\sigma$ of discrete Gaussian distribution is small, then any vector extracted from this distribution will possibly be short.*

**Lemma 4.** *Let $m, n, q$ denote as the positive integers that satisfy $m > n$ and $q > 2$. We define a matrix $A \in \mathbb{Z}_q^{n \times m}$ with parameter $\sigma = \widetilde{\Omega}(n)$ and $\mathcal{L} = \lambda_q^\perp(A)$. Then, there is a negligible function **negl**($\cdot$) such that:*

$$\Pr_{x \leftarrow \mathcal{D}_{\mathcal{L}, \sigma}} \left[ \|x\| > \sqrt{m}\sigma \right] \leq \textbf{negl}(n) \tag{5}$$

*where $\|x\|$ is the $\ell_2$ norm of $x$.*

2.4.2. Truncated Discrete Gaussians

The truncated discrete Gaussian distribution $\widetilde{\mathcal{D}}_{\mathbb{Z}^m, \sigma}$ with the parameter $\sigma$ over $\mathbb{Z}^m$ is the same as the discrete Gaussian distribution $\mathcal{D}_{\mathbb{Z}^m, \sigma}$. Expect that it outputs 0 when the $\ell_\infty$ norm is larger than $\sqrt{m}\sigma$. In addition, according to Lemma 1, $\widetilde{\mathcal{D}}_{\mathbb{Z}^m, \sigma}$ and $\mathcal{D}_{\mathbb{Z}^m, \sigma}$ are statistically indistinguishable.

2.4.3. Lattice Trapdoors

Lattices with trapdoors function have certain "trapdoors" that allow efficient solutions to hard lattice problems. A trapdoor lattice consists of the following two algorithms:

① **TrapGen**$(1^n, 1^m, q) \mapsto (A, T_A)$: The lattice generation algorithm is a randomized algorithm. It takes as input the dimensions $n, m$ and modulus $q$ of the matrix and output a matrix $A \in \mathbb{Z}_q^{n \times m}$ together with a lattice trapdoors function $T_A$.

② **SamplePre**$(A, T_A, \sigma, u) \mapsto s$: The pre-sampling algorithm takes a matrix $A$, the lattice trapdoors function $T_A$, a vector $u \in \mathbb{Z}_q^n$, and a parameter $\sigma \in \mathbb{R}$ as the inputs. And it outputs the vector $s \in \mathbb{Z}_q^m$, which the vector $s$ satisfies $A \cdot s^\top = u^\top \ \|s\| \leq \sqrt{m} \cdot \sigma$.

### 2.5. Learning with Errors (LWE)

For a security parameter $\lambda \in \mathbb{N}$, assuming that $n : \mathbb{N} \to \mathbb{N}$, $q : \mathbb{N} \to \mathbb{N}$ and $\sigma : \mathbb{N} \to \mathbb{R}^+$ are the function of $\lambda$. We define $\text{LWE}_{n,q,\sigma}$ as the hypothesis of the LWE-hardness problem by parametric $q = q(\lambda)$, $n = n(\lambda)$ and $\sigma = \sigma(\lambda)$. For any PPT adversary $\mathcal{A}$, there exists a negligible function $\boldsymbol{negl}(\cdot)$ for any $\lambda \in \mathbb{N}$:

$$\text{Adv}_{\mathcal{A}}^{\text{LWE}_{n,q,\sigma}}(\lambda) \triangleq \left| \Pr \left[ \begin{array}{c} 1 \leftarrow \mathcal{A}^{\mathcal{O}_1^s(\cdot)}(1^\lambda) \\ s \leftarrow \mathbb{Z}_q^n \end{array} \right] \atop -\Pr \left[ 1 \leftarrow \mathcal{A}^{\mathcal{O}_2(\cdot)}(1^\lambda) \right] \right| \leq \boldsymbol{negl}(\lambda) \tag{6}$$

where the Oracles $\mathcal{O}_1^s(\cdot)$ and $\mathcal{O}_2(\cdot)$ are defined as follows:

$\mathcal{O}_1^s(\cdot)$ is strongly connected with $s \in \mathbb{Z}_q^n$, and it chooses $a \leftarrow \mathbb{Z}_q^n$ and $e \leftarrow \mathcal{D}_{\mathbb{Z},\sigma}$, and outputs $(a, sa^\top + e \bmod q)$ in each query. $\mathcal{O}_2(\cdot)$ on each query it chooses $a \leftarrow \mathbb{Z}_q^n$ and $u \leftarrow \mathbb{Z}_q$, and outputs $(a, u)$.

**Definition 1.** *If there is a hypothesis that a PPT adversary can break the LWE assumption, then there exists a PPT quantum algorithm that can solve some hard lattice problems in the worst case.*

Given the current technical schemes on the hard lattice problems, the LWE assumption is believed to be true for any polynomial $n(\cdot)$ and function $q(\cdot)$ when all $\lambda \in \mathbb{N}$, $n = n(\lambda)$, $q = q(\lambda)$, and $\sigma = \sigma(\lambda)$ satisfy the following conditions:

$$2\sqrt{n} < \sigma < q < 2^n, \; n \cdot q / \sigma < 2^{n^\epsilon}, \; 0 < \epsilon < 1/2 \tag{7}$$

## 3. LWE-CP-ABE Scheme

### 3.1. Algorithm Construction

In this section, we present the LWE-CP-ABE scheme for access structures with LSSS using blockchain technology, which is associated with Proxy Re-Encryption (PRE) protocol under a subset of on-chain node client by transforming encrypted data from one public key to another. As shown in Figure 1, our CP-ABE system under LWE assumption LWE – CP – ABE = (Setup, KeyGen, Enc, Dec, AccGen, AccUpdate) mainly consists of six procedures with the following syntax:

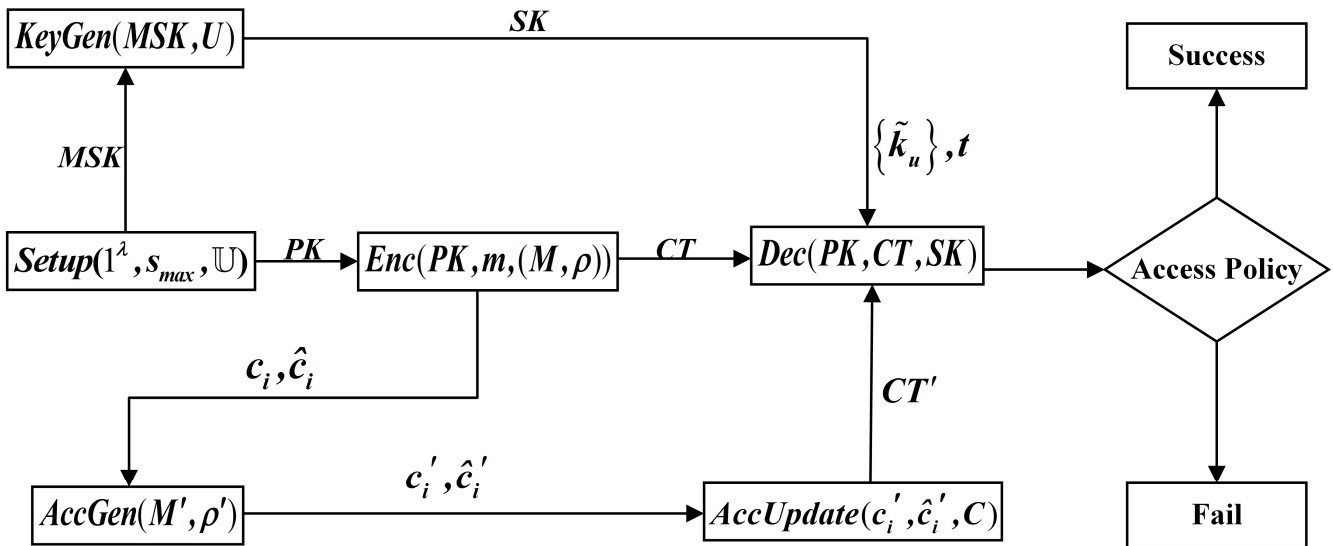

**Figure 1.** Flowchart of LWE-CPABE.

(1) **Setup**$(1^\lambda, s_{\max}, \mathbb{U}) \to (\text{PK}, \text{MSK})$

The setup algorithm takes as input the security parameter $\lambda$, the maximum width $s_{\max} = s_{\max}(\lambda)$ supported by an LSSS matrix supported by the scheme, and the attribute universe $\mathbb{U}$.

For each attribute $u$ in the system, it selects $A_u \in \mathbb{Z}_q^{n \times m}$ to generate the trapdoors function $T_{A_u}$, and selects the uniformly distributed random matrix $H_u \leftarrow \mathbb{Z}_q^{n \times m}$ and random vector $y \leftarrow \mathbb{Z}_q^n$. It outputs the system's public parameters and the master secret key.

$$\text{PK} = (y, \{A_u\}, \{H_u\}), \ \text{MSK} = \{T_{A_u}\} \tag{8}$$

(2) **KeyGen**$(\text{MSK}, U) \to \text{SK}$

The key generation algorithm takes as input the user's attribute set $U$ and MSK. We let a vector $\hat{t} \leftarrow \text{noise}^{m-1}$ and set the vector $t = (1, \hat{t}) \in \mathbb{Z}^m$, The vector $t$ is a part of the user attribute SK, and every user has a different $t$, which can prevent conspiracy attacks. For each attribute $u \in U$, a short vector $\widetilde{k}_u$ is generated by the trapdoors $T_{A_u}$ and satisfies $A_u \widetilde{k}_u^\top = H_u t^\top$. Finally, it outputs:

$$\text{SK} = \left( \left\{ \widetilde{k}_u \right\}, t \right) \tag{9}$$

(3) **Enc**$(\text{PK}, m, (M, \rho)) \to \text{CT}$

The encryption algorithm takes as input the public parameter PK, a message bit $\mathbf{m} \in \{\mathbf{0}, \mathbf{1}\}$ and access policy $(M, \rho)$ from LSSS. Let $\rho$ be the mapping function that maps the attribute to the row of the matrix $M$. that is, $\rho(i)$ is the attribute $M = (M_{i,j})_{l \times s_{max}} \in \{-1, 0, 1\}^{l \times s_{max}} \subset \mathbb{Z}_q^{l \times s_{max}}$ associated with the $i$ row in the matrix $M$. The procedure Randomly samples vectors $s \leftarrow \mathbb{Z}_q^n$, $v_2, \cdots, v_{s_{\max}} \leftarrow \mathbb{Z}_q^m$ and $\{x_i\} \leftarrow \mathbb{Z}_q^n$, and then it computes as follows:

$$\begin{aligned} c_i &= sA_{\rho(i)} + \text{noise} \\ \hat{c}_i &= M_{i,1} \left( sy^\top, \overbrace{0, \cdots, 0}^{m-1} \right) + \left[ \sum_{j \in \{2, \cdots, s_{\max}\}} M_{i,j} v_j \right] \\ &\quad - sH_{\rho(i)} + \text{noise} \end{aligned} \tag{10}$$

and it outputs:

$$\text{CT} = \left( \{c_i\}_{i \in [\ell]}, \{\hat{c}_i\}_{i \in [\ell]}, C = \text{MSB}\left( sy^\top \right) \oplus m \right) \tag{11}$$

$$\text{SK}_{GID,u} = \hat{k}_{GID,u} + \hat{k}_{GID,u} \tag{12}$$

(4) **Dec**$(\text{PK}, \text{CT}, \text{SK}) \to m$

The decryption algorithm takes as inputs the PK, the ciphertext CT generated with respect to an LSSS, and the user secret key **SK** related to some subset of attributes U. Let the attribute owned by the user meet the access control policy. We denote $I$ as the row vector set corresponding to the attribute and $\{\omega_i\}_{i \in I} \in \{0, 1\} \subset \mathbb{Z}_q$ the reconstruction coefficient. For any $i \in I$, let $\rho(i)$ be the row-relevant attribute. The algorithm computes:

$$K = \sum_{i \in I} \omega_i \left( c_i \widetilde{k}_{\rho(i)}^\top + \hat{c}_i t^\top \right) \tag{13}$$

and outputs:

$$m = C \oplus \text{MSB}(K) \tag{14}$$

(6) **AccGen**$(M', \rho') \to (c_i', \hat{c}_i')$.

The ciphertext policy generation algorithm takes as input the new access control policy and outputs the updated policy ciphertext.

(7) **AccUpdate**$\left(c_i', \hat{c}_i', C\right) \to \mathrm{CT}'$.

The ciphertext policy update algorithm takes the policy ciphertext generated by the policy generation algorithm as the input and outputs the new ciphertext $\mathrm{CT}'$.

*3.2. Security Model*

The proposed CP-ABE scheme is selectively secure under linear independence restriction in the random oracle model if the LWE assumption holds. To prove this assumption, the hybrid security game contains a challenger and an adversary. The challenger simulates the system and answers the adversary's inquiries. The hybrid game starts as follows.

(1) Setup phase. The reduction algorithm receives a security parameter $1^\lambda$ and an access control policy $(M, \rho)$ from the challenger and invokes an adversary. The challenger runs the **Setup** algorithm to generate the system **PK** and send it to the adversary.

(2) Secret key queried by an adversary. For each key query, the adversary sends a set of attributes $U \in \mathbb{U}$, but these attributes do not satisfy the access control policy $(M, \rho)$. The challenger runs the **KeyGen** algorithm and sends the generated user attribute SK to the adversary.

(3) Challenge phase. To generate the challenge ciphertext, the challenger selects a random bit $b \leftarrow \{0, 1\}$ and runs the **ENC** algorithm to encrypt it using the access control policy. Consequently, the challenger provides that to the adversary.

(4) Repeat step (2).

(5) Guess phase. The hybrid game terminates with an adversary outputting its guess $b' \leftarrow \{0, 1\}$ for the bit $b$ encrypted within the challenge ciphertext.

The advantage of the adversary $\mathcal{A}$ in this game is:

$$\mathrm{Adv}_{\mathcal{A}}^{\mathrm{LWE-CPABE,\ SEL-CPA}}(\lambda) \triangleq \left| \Pr[b = b'] - 1/2 \right| \tag{15}$$

**Definition 2.** *If there exists a negligible function* **negl**$(\cdot)$ *for any PPT adversary* $\mathcal{A}$, *for all* $\lambda \in \mathbb{N}$, *there is* $\mathrm{Adv}_{\mathcal{A}}^{\mathrm{LWE-CPABE,SEL-CPA}}(\lambda) \leq$ **negl**$(\lambda)$. *The LWE-CP-ABE scheme proposed in this paper is selectively secure under linear independence restriction.*

## 4. Blockchain-Based LWE-CP-ABE Data Sharing Scheme

*4.1. Overview of Blockchain-Based LWE-CP-ABE*

To ensure efficient data sharing and access policy updating on the blockchain, all activities of the ciphertext access control scheme, such as generating the public parameters, secret keys, and oracle functions, are written into the distributed ledger on blockchain. Users can access the authorized data through their user attribute secret key controlled over the blockchain, which matches their attributes, and complete the secure and traceable data sharing. The blockchain-based data-sharing framework of the LWE-CP-ABE is shown in Figure 2. Here, the blockchain system is built with pre-quantum cryptography for master key management and immutable storage of transactions, and CP-ABE is constructed with post-quantum cryptography for access and sharing of ciphertext. The system characters play particular roles described as follows:

(1) User (DO, DU). It includes the data owner (DO) and the data user (DU).

When a DO wants to share this information with DU, they can create a policy with the LSSS matrix to grant access to DU. Thus DO generates the global parameters, encrypts the corresponding message, and uploads the transactions to the blockchain.

At the same time, if different DUs want to access the data, they must request the secret keys from the nodes on the blockchain. The DUs use it to decrypt the ciphertext obtained from the third-party storage;

(2) Third-party storage (TPS). It provides API storage services for encrypted data, such as distributed file-sharing network IPFS, and its hash addresses are stored on the blockchain;

(3) Blockchain Network (BN). It comprises a participant node on the blockchain that stands ready to provide decentralized key management services and creates a symmetric key for the master secret key of CP-ABE for the data users. Additionally, the blockchain

system provides traceability and auditability of transactions involved in the LWE-CP-ABE access control protocol;

(4) Key Management Network (KMN). A group of blockchain nodes that provides an application of the Proxy Re-Encryption (PRE) scheme ensuring the security of the master secret key distribution in the LWE-CPABE encryption algorithm.

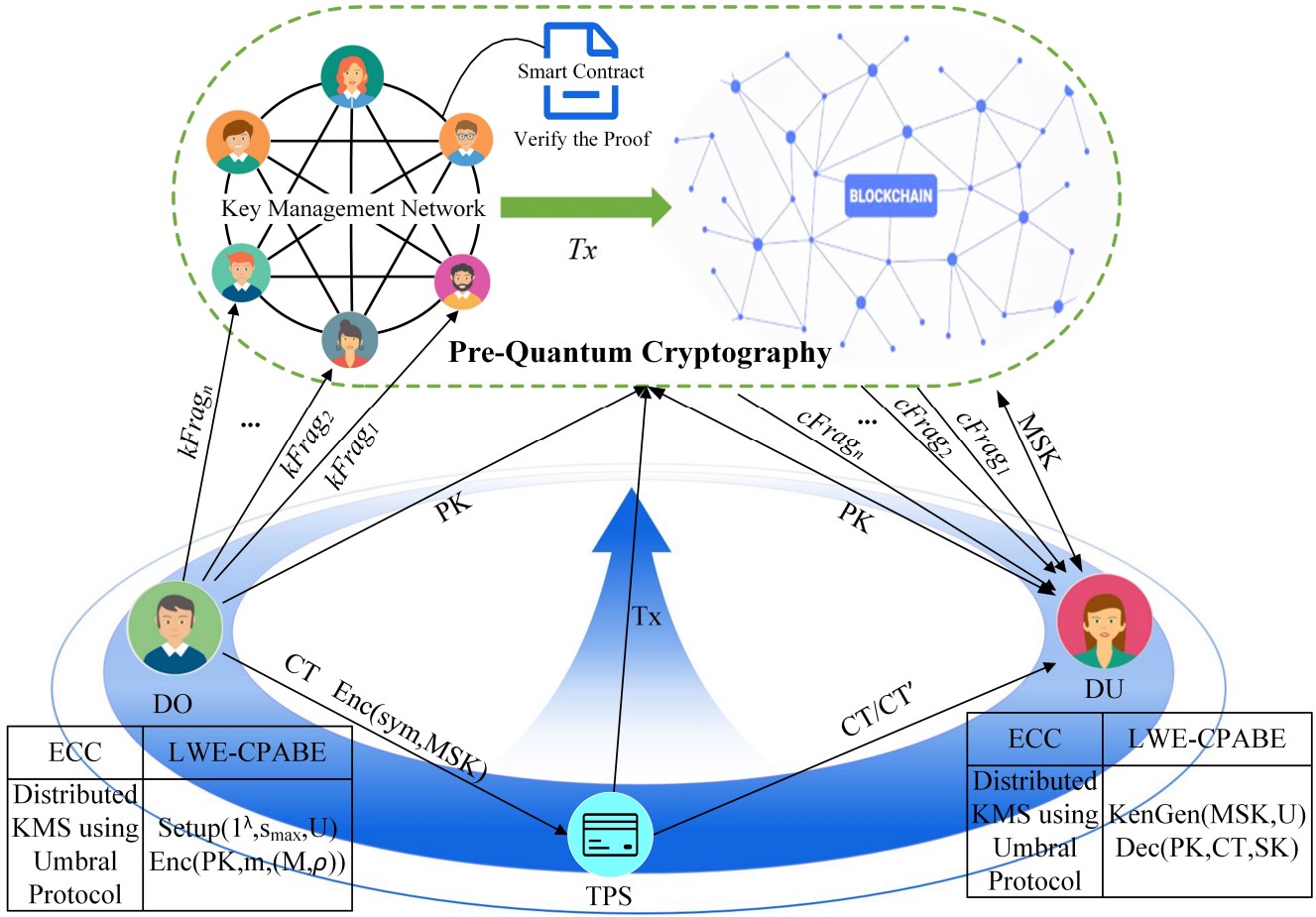

**Figure 2.** System architecture.

### 4.2. Data Sharing Scheme in Key Management Network and Blockchain Network

In our LWE-CP-ABE approach based on blockchain for data access and sharing, it is essential that DU can securely acquire the master secret key MSK without needing to involve a central authority or trusted third party. Thus, we introduce the PRE algorithm with ECC applied in the blockchain network, which acts as a decentralized key management service to carry out the master key distribution for data users. To grant the MSK to DU, DO creates a random symmetric key, encrypts the MSK for DUs, and each node is responsible for securely storing and managing the re-encryption keys or the key shares. DU must collect the fragments until he obtains a threshold, $t$, number of fragments to re-constructed and decrypt the ciphertext at the DU side. The mathematical details of the LWE-CP-ABE protocol running in KMN and BN are described as follows.

In the LWE-CP-ABE scheme, the algorithms select the parameters $n, m, \sigma, q$ and the noise distributions $\mathcal{X}_{lwe}$, $\mathcal{X}_1$, $\mathcal{X}_2$, $\mathcal{X}_{big}$. For any $B \in \mathbb{N}$, the notation $\mathcal{U}_B$ represents the uniform distribution $\mathbb{Z} \cap [-B, B]$.

$$
\begin{aligned}
&n = \text{play}(\lambda), \sigma < q, n \cdot q / \sigma < 2^{n^\varepsilon}, \mathcal{X}_{lwe} = \widetilde{\mathcal{D}}_{\mathbb{Z},\sigma} (\textit{for LWE security}) \\
&m > 2s_{\max} n \log q + \omega \log n + 2\lambda (\textit{for enhanced trapdoor sampling and LHL}) \\
&\sigma > \sqrt{s_{\max} n \log q \log m} + \lambda (\textit{for enhanced trapdoor sampling}) \\
&\mathcal{X}_1 = \widetilde{\mathcal{D}}_{\mathbb{Z}^{n-1},\sigma}, \mathcal{X}_2 = \widetilde{\mathcal{D}}_{\mathbb{Z}^n,\sigma} (\textit{for enhanced trapdoor sampling}) \\
&\mathcal{X}_{big} = \mathcal{U}_{\hat{B}}, \hat{B} > (m^{3/2}\sigma + 1)2^\lambda (\textit{for smudging/security}) \\
&\left|\mathbb{U}\right| \cdot 3m^{3/2}\sigma\hat{B} < q/4 (\textit{for correctness})
\end{aligned}
\tag{16}
$$

(1) System setup. The DO selects the security parameter $\lambda$ encoded in unary, the maximum width $s_{\max}$ supported by the LSSS matrix, and the user attribute set $\mathbb{U}$ associated with the system. The **Setup** algorithm (Algorithm 1) generates the public key PK and the master secret key MSK.

---

**Algorithm 1: Setup**

---

Input: the security parameter $\lambda$, the maximum width $s_{\max}$ of an LSSS matrix, and the user attribute set supported by the system;

Output: the PK and the MSK,

1. *Choose an LWE modulus $q$, dimensions $n$, $m$ and distributions $\mathcal{X}_{lwe}$, $\mathcal{X}_1$, $\mathcal{X}_2$, $\mathcal{X}_{big}$;*
2. *Choose a vector $\mathbf{y} \leftarrow \mathbb{Z}_q^n$ and a matrices $\{\mathbf{H}_u\}_{u \in \mathbb{U}} \leftarrow \mathbb{Z}_q^{n \times m}$;*
3. **EnTrapGen**$(1^n, 1^m, q) \rightarrow \{(\mathbf{A}_u, T_{\mathbf{A}_u})\}$ /* trapdoors computation */

$$
\text{PK} = \begin{pmatrix} n, m, q, \mathcal{X}_{lwe}, \mathcal{X}_1, \mathcal{X}_2, \mathcal{X}_{big}, \mathbf{y}, \\ \{\mathbf{A}_u\}_{u \in \mathbb{U}}, \{\mathbf{H}_u\}_{u \in \mathbb{U}} \end{pmatrix}
$$

$$
\text{MSK} = \{T_{\mathbf{A}_u}\}_{u \in \mathbb{U}}
$$

---

The DO sends the transaction and uploads the PK on the blockchain BN. Specifically, all the public information is recorded in the block through the transaction $Tx_{UploadPK}$ = {*DO, BN, A, P, Timestamp, PK, Sig$_{DO}$, $Coin*}, where *A* is the LSSS matrix of access control policy, and *P* represents the operation of a publishing contract, *Timestamp* is the time stamp of the transaction, *Sig$_{DO}$* is the signature of DO, and *$Coin* is the payment for transaction fee.

In addition, DO has the MSK they want to share with the DUs. DO can create a PRE protocol running on the nodes of KMN to grant access to DUs. Therefore, the KMN system still uses pre-quantum cryptography with a cyclic group $\mathbb{G}$ of prime order $q$. Let $g \in \mathbb{G}$ be the generator. Then, DO generates a pair of public and secret keys in the blockchain system. That is, sample $a \in \mathbb{Z}_q$ uniformly at random, computes $g^a$ and outputs the key pair (*pk, sk*) = ($g^a$, *a*). The re-encryption key generation algorithm takes as input the DO's secret key *a*, the DU's public key, $pk_{DU} = g^b$, several fragments N, and a threshold *t*, DO computes N fragments of the re-encryption key $rk_{DO \rightarrow DU}$ between DO and DU, each of them named *kFrag*. In this step, an ephemeral symmetric key generated by DO is used to encrypt the MSK, and the symmetric key is encrypted using DO's asymmetric encryption key. The encrypted MSK and the encrypted symmetric key (the KEM portion, called a capsule) are stored together in TPS. Next, as shown in Figure 3, to gain the MSK, DU requests the capsule-associated key fragments *kFrag* to give to the KMN for re-encryption operation. Those *n* designated nodes in KMN perform a partial operation to produce a corresponding ciphertext fragment named *cFrag*. Meanwhile, a correctness proof $\pi$ for the resulting fragment is constructed using a non-interactive zero-knowledge proof of discrete logarithm equality to verify the correctness. These fragments are returned to DU, which collects *cFrag*s until it obtains an *m-of-n* threshold. Finally, DU can decrypt the capsule to obtain the symmetric key *K* to decrypt the encrypted MSK.

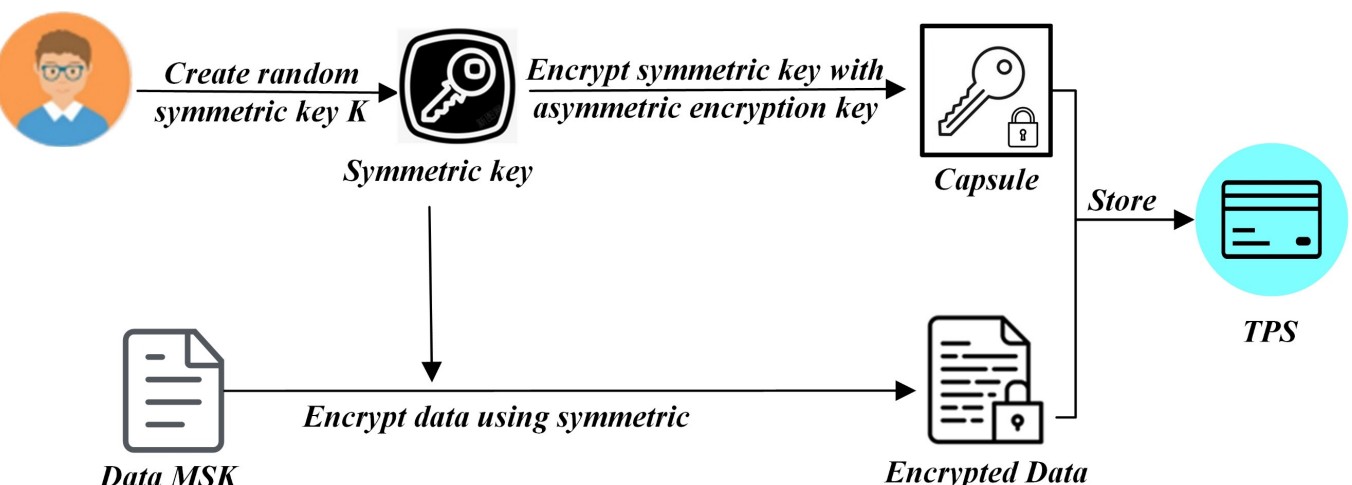

**Figure 3.** Proxy Re-Encryption approach for MSK.

Thus, to keep this paper compact, all above mentioned PRE protocol that we introduced uses the UMBERAL scheme [43] and is served as a decentralized key management system to distribute and manage the MSK that is applied to key generation and decryption for DU of LWE-CP-ABE. Meanwhile, the activities of DO/DU, including authorization, request, and policy making, are written into blockchain by a smart contract.

(2) Data encryption. The DO formulates the access control policy $(M, \rho)$. $(M, \rho)$ is the LSSS access control policy, in which $M = (M_{i,j})_{\ell \times s_{\max}} \in \{-1, 0, 1\}^{\ell \times s_{\max}} \subset \mathbb{Z}_q^{\ell \times s_{\max}}$, $\rho$ represents the mapping (monomorphism) function $\rho : [\ell] \to \mathbb{U}$, and it maps the access control matrix $M_i$ to the attribute set $\mathbb{U}$. After that, the DO runs the **Enc** algorithm, inputs the PK, the plaintext $m$, and the access control policy $(M, \rho)$, and gets the CT.

The specific algorithm is shown in Algorithm 2.

---

**Algorithm 2:** Data Encryption Algorithm

---

Input: the PK, the plaintext $m$, and the access control policy $(M, \rho)$;
Output: the CT.

1. *Generate Access Policy $(M, \rho)$;*
2. *Select a random vector $s \leftarrow \mathbb{Z}_q^n$; /\* the secret sharing key \*/*
3. *Sample a vector $\{v_j\}_{j \in \{2, \cdots, s_{\max}\}} \leftarrow \mathbb{Z}_q^m$;*
4. for each $i \in [\ell]$
    *Select random noise $\{e_i\}_{i \in [\ell]} \leftarrow \mathcal{X}_{lwe}^m$ and $\{\hat{e}_i\}_{i \in [\ell]} \leftarrow \mathcal{X}_{big}^m$;*
    Compute $c_i, \hat{c}_i \in \mathbb{Z}_q^m$
We have;

$$c_i = sA_{\rho(i)} + e_i$$

$$\hat{c}_i = M_{i,1}\left(sy^\top, \overbrace{0, \cdots, 0}^{m-1}\right) + \left[\sum_{j \in \{2, \cdots, s_{\max}\}} M_{i,j}v_j\right]$$

$$- sH_{\rho(i)} + \hat{e}_i$$

5. $\mathrm{CT} = \left((M, \rho), \{c_i\}_{i \in [\ell]}, \{\hat{c}_i\}_{i \in [\ell]}, C = \mathrm{MSB}\left(sy^\top\right) \oplus m\right).$

---

The DO uploads the CT to the TPS, which generates the contract transaction $Tx_{CTAddress} = \{TPS, BN, A, D, Timestamp, CT_{Address}, Sig_{TPS}, \$Coin\}$ to upload the hash address of CT on the blockchain. In this transaction, the parameter $D$ denotes the operation type is a data message.

(3) User attribute secret key generation. Each DU firstly requires m of these interactions with m different nodes to obtain a fully re-encrypted capsule. Secondly, DU combines the

fragments to decrypt the re-encrypted capsule using his private key. Finally, DU obtains the symmetric key to decrypt encrypted MSK.

DU inputs the attribute information $U$ and the MSK and runs the **KeyGen** algorithm to output the user attribute $SK_{GID}$ corresponding to the attribute. The specific algorithm is shown in Algorithm 3.

---

**Algorithm 3:** User Attribute Secret Key Generation Algorithm

---

Input: the MSK and own attribute information $U$;
Output: the user attribute SK,

1. *Select a random vector* $\hat{t} \leftarrow \mathcal{X}_1$;
2. $t = (1, \hat{t}) \in \mathbb{Z}^m$;
3. **for each** $u \in U$

   $\quad\quad$ *Select* $\hat{k}_u \leftarrow \mathcal{X}_{big}^m$;

   $\quad$ *Compute* $\begin{array}{l} \widetilde{k}_u \leftarrow \textbf{EnSamplePre} \\ \left(A_u, T_{A_u}, \sigma, tH_u^\top - \hat{k}_u A_u^\top\right) \end{array}$

   **then** *Compute* $k_u = \hat{k}_u + \widetilde{k}_u$;
   $\quad$ Output the user attribute $SK_{GID}$:

   $$SK = (\{k_u\}_{u \in U}, t)$$

---

(4) Ciphertext decryption. When DU wants to access the data, he retrieves the ciphertext address from the blockchain BN, searches the corresponding CT from the TPS through the ciphertext address $CT_{Address}$ and downloads it to the local through the transaction $Tx_{DownloadCT} = \{TPS, DU, D, P, Timestamp, CT, Sig_{TPS}, \$Coin\}$, where $P$ denotes the trading type as publishing contract. The DU uses the SK to decrypt the CT and obtain the plaintext.

During the process of ciphertext decryption, the SK corresponds to a certain subset $U \in \mathbb{U}$ of the attribute set $\mathbb{U}$. If $(1, 0, \cdots, 0)$ is not in the row space of the matrix $M$ associated with $U$, then the decryption fails. Otherwise, let $I$ be a set of row indexes of the matrix $M$ and satisfy $\forall i \in I : \rho(i) \in U$. Let $\{\omega_i\}_{i \in I} \in \{0, 1\} \subset \mathbb{Z}_q$ and $\sum_{i \in I} \omega_i M_i = (1, 0, \cdots, 0)$ be scalar, where $M_i$ is the row $i$ of the matrix $M$. The specific algorithm is shown in Algorithm 4.

---

**Algorithm 4:** Decryption Algorithm

---

Input: the CT and the user attribute $SK_{GID}$;
Output: the plaintext $m$.

1. *Compute* $K = \sum_{i \in I} \omega_i \left(c_i k_{\rho(i)}^\top + \hat{c}_i t^\top\right)$
2. *Compute* $m = C \oplus \text{MSB}(K)$;

---

(5) Ciphertext policy generation. When the attribute set or access control policy needs to be changed, the DU can update the access control policy of the original ciphertext retention. To update the policy, the DO generates a new access control policy $(M', \rho')$, and runs the **AccGen** algorithm to get the updated policy ciphertext $Update_{CT} = (c_i{}', \hat{c}_i')$:

$$
\begin{aligned}
c_i{}' &= sA_{\rho'(i)} + e_i \\
\hat{c}_i' &= M'_{i,1} \left( sy^\top, \overbrace{0, \cdots, 0}^{m-1} \right) + \left[ \sum_{j \in \{2, \cdots, s_{\max}\}} M'_{i,j} v_j \right] \\
&\quad - sH_{\rho(i)} + \hat{e}_i
\end{aligned}
\tag{17}
$$

At the same time, the DO sends transaction $Tx_{AccGen} = \{DO, TPS, A, N, Timestamp, Update_{CT}, Sig_{DO}, \$Coin\}$ to record the updating operation representing $N$ on the blockchain. The DO store the ciphertext of the access policy on TPS, and its hash value is written in the transaction.

(6) Ciphertext policy update. When DO obtains the ciphertext C derived from the original CT, the updated access control policy ciphertext $(c_i', \hat{c}_i')$ is used to run the **AccUpdate** algorithm to generate the new ciphertext CT′, which is still uploaded to the original ciphertext address to facilitate the decryption procedures for data users.

$$\text{CT}' = \begin{pmatrix} (\boldsymbol{M}',\rho'), \{\boldsymbol{c_i}'\}_{i\in[\ell]}, \{\hat{\boldsymbol{c}}_i'\}_{i\in[\ell]'} \\ C = \text{MSB}(\boldsymbol{sy}^\top) \oplus m \end{pmatrix} \tag{18}$$

To an extent, the users can quickly and efficiently retrieve the required information using the formatted transaction structure in the blockchain encryption protocol based on the LWE-CP-ABE. Meanwhile, the whole log of each node's process event is recorded into the ledger on blockchain BN, which can offer better auditability and accountability in the dynamic access control system.

## 5. Analysis of the Scheme

### 5.1. Analysis of Correctness

Assuming that a data user has the attributes $\boldsymbol{u} \in \mathbb{U}$ and the LSSS access control policy $(\boldsymbol{M}, \rho)$ for which $\boldsymbol{U}$ constitute an authorized set. By construction,

$$K = \sum_{i\in\boldsymbol{I}} \omega_i \left( \boldsymbol{c_i k}_{\rho(i)}^\top + \hat{\boldsymbol{c}}_i \boldsymbol{t}^\top \right) \tag{19}$$

Expanding $\{\boldsymbol{c_i}\}_{i\in\boldsymbol{I}}$ and $\{\hat{\boldsymbol{c}}_i\}_{i\in\boldsymbol{I}'}$ we get

$$
\begin{aligned}
K = &\sum_{i\in\boldsymbol{I}} \omega_i \boldsymbol{s} \boldsymbol{A}_{\rho(i)} \boldsymbol{k}_{\rho(i)}^\top + \sum_{i\in\boldsymbol{I}} \omega_i M_{i,1} (\boldsymbol{sy}^\top, 0, \cdots, 0) \boldsymbol{t}^\top \\
&+ \sum_{i\in\boldsymbol{I}, j\in\{2,\cdots,s_{\max}\}} \omega_i M_{i,j} \boldsymbol{v}_j \boldsymbol{t}^\top - \sum_{i\in\boldsymbol{I}} \omega_i \boldsymbol{s} \boldsymbol{H}_{\rho(i)} \boldsymbol{t}^\top \\
&+ \sum_{i\in\boldsymbol{I}} \omega_i \hat{\boldsymbol{e}}_i \boldsymbol{t}^\top
\end{aligned}
\tag{20}
$$

For each $u \in \boldsymbol{U}$, we run **EnSamplePre** algorithm, and then $\boldsymbol{A}_u \widetilde{\boldsymbol{k}}_u^\top = \boldsymbol{H}_u \boldsymbol{t}^\top - \boldsymbol{A}_u \hat{\boldsymbol{k}}_u^\top$ Therefore, for each $i \in \boldsymbol{I}$, it holds that $\boldsymbol{A}_{\rho(i)} \boldsymbol{k}_{\rho(i)}^\top = \boldsymbol{A}_{\rho(i)} \hat{\boldsymbol{k}}_{\rho(i)}^\top + \boldsymbol{A}_{\rho(i)} \widetilde{\boldsymbol{k}}_{\rho(i)}^\top = \boldsymbol{H}_{\rho(i)} \boldsymbol{t}^\top$, Hence,

$$
\begin{aligned}
K = &\sum_{i\in\boldsymbol{I}} \omega_i \boldsymbol{s} \boldsymbol{H}_{\rho(i)} \boldsymbol{t}^\top + \sum_{i\in\boldsymbol{I}} \omega_i M_{i,1} (\boldsymbol{sy}^\top, 0, \cdots, 0) \boldsymbol{t}^\top \\
&+ \sum_{i\in\boldsymbol{I}, j\in\{2,\cdots,s_{\max}\}} \omega_i M_{i,j} \boldsymbol{v}_j \boldsymbol{t}^\top - \sum_{i\in\boldsymbol{I}} \omega_i \boldsymbol{s} \boldsymbol{H}_{\rho(i)} \boldsymbol{t}^\top \\
&+ \sum_{i\in\boldsymbol{I}} \omega_i \boldsymbol{e}_i \boldsymbol{k}_{\rho(i)}^\top + \sum_{i\in\boldsymbol{I}} \omega_i \hat{\boldsymbol{e}}_i \boldsymbol{t}^\top \\
= &\sum_{i\in\boldsymbol{I}} \omega_i M_{i,1} (\boldsymbol{sy}^\top, 0, \cdots, 0) \boldsymbol{t}^\top \\
&+ \sum_{i\in\boldsymbol{I}, j\in\{2,\cdots,s_{\max}\}} \omega_i M_{i,j} \boldsymbol{v}_j \boldsymbol{t}^\top + \sum_{i\in\boldsymbol{I}} \omega_i \boldsymbol{e}_i \boldsymbol{k}_{\rho(i)}^\top \\
&+ \sum_{i\in\boldsymbol{I}} \omega_i \hat{\boldsymbol{e}}_i \boldsymbol{t}^\top \\
= &\left( \sum_{i\in\boldsymbol{I}} \omega_i M_{i,1} \right) (\boldsymbol{sy}^\top, 0, \cdots, 0) \boldsymbol{t}^\top \\
&+ \sum_{i\in\boldsymbol{I}, j\in\{2,\cdots,s_{\max}\}} \omega_i M_{i,j} \boldsymbol{v}_j \boldsymbol{t}^\top + \sum_{i\in\boldsymbol{I}} \omega_i \boldsymbol{e}_i \boldsymbol{k}_{\rho(i)}^\top \\
&+ \sum_{i\in\boldsymbol{I}} \omega_i \hat{\boldsymbol{e}}_i \boldsymbol{t}^\top
\end{aligned}
\tag{21}
$$

When $\sum_{i \in I} \omega_i M_{i,j} = 1$ for $1 < j \leq s_{\max}$, it holds that $\sum_{i \in I} \omega_i M_{i,j} = 0$. Also, $\boldsymbol{t} = (1, \hat{\boldsymbol{t}})$ is constructed using **KeyGen** in Section 3.1, and hence $(\boldsymbol{sy}^\top, 0, \cdots, 0) \boldsymbol{t}^\top = \boldsymbol{sy}^\top$. Thus,

$$K = \boldsymbol{sy}^\top + \sum_{i \in I} \omega_i \boldsymbol{e}_i \boldsymbol{k}_{\rho(i)}^\top + \sum_{i \in I} \omega_i \hat{\boldsymbol{e}}_i \boldsymbol{t}^\top \tag{22}$$

As for the noise part $\sum_{i \in I} \omega_i \boldsymbol{e}_i \boldsymbol{k}_{\rho(i)}^\top + \sum_{i \in I} \omega_i \hat{\boldsymbol{e}}_i \boldsymbol{t}^\top$, the following inequalities hold except with negligible probability.

(1) According to Lemma 2, the positive integer $m$ coordinates in $\boldsymbol{e}_i$ are from truncated discrete Gaussians distribution $\widetilde{\mathcal{D}}_{\mathbb{Z},\sigma}$, so there is $\|\boldsymbol{e}_i\| \leq \sqrt{m}\sigma$.

(2) the positive integer $m$ coordinates in $\hat{\boldsymbol{e}}_i$ are from the uniform distribution $\mathbb{Z} \cap [-\hat{B}, \hat{B}]$, so there is $\|\hat{\boldsymbol{e}}_i\| \leq \sqrt{m}\hat{B}$.

(3) $m$ coordinates in $\hat{\boldsymbol{k}}_{\rho(i)}$ are from the uniform distribution $\mathbb{Z} \cap [-\hat{B}, \hat{B}]$, so there is $\|\hat{\boldsymbol{k}}_{\rho(i)}\| \leq \sqrt{m}\hat{B}$. And $m$ coordinates in $\widetilde{\boldsymbol{k}}_{\rho(i)}$ are statistically close to the truncated discrete Gaussians distribution $\widetilde{\mathcal{D}}_{\mathbb{Z}^m,\sigma}$, so there is $\|\widetilde{\boldsymbol{k}}_{\rho(i)}\| \leq m\sigma$. To sum up, for $\|\boldsymbol{k}_{\rho(i)}\|$, $\boldsymbol{k}_{\rho(i)} = \hat{\boldsymbol{k}}_{\rho(i)} + \widetilde{\boldsymbol{k}}_{\rho(i)}$, so the boundary on $\|\boldsymbol{k}_{\rho(i)}\|$ is $\|\boldsymbol{k}_{\rho(i)}\| \leq m\sigma + \sqrt{m}\hat{B}$.

(4) if $\boldsymbol{t} = (1, \hat{\boldsymbol{t}})$, where $\hat{\boldsymbol{t}}$ comes from a truncated discrete Gaussians distribution $\widetilde{\mathcal{D}}_{\mathbb{Z}^{m-1},\sigma}$, then it holds $\|\boldsymbol{t}\| < m\sigma$.

Therefore, given the above conditions, we have that:

$$\begin{aligned}
&\left\| \sum_{i \in I} \omega_i \boldsymbol{e}_i \boldsymbol{k}_{\rho(i)}^\top + \sum_{i \in I} \omega_i \hat{\boldsymbol{e}}_i \boldsymbol{t}^\top \right\| \\
&< |\mathbb{U}| \left( m^{3/2}\sigma^2 + m\sigma\hat{B} + m^{3/2}\sigma\hat{B} \right) \\
&< |\mathbb{U}| \cdot 3m^{3/2}\sigma\hat{B} \\
&< q/4
\end{aligned} \tag{23}$$

Therefore, with almost negligible probability in $\lambda$, the MSB of $sy^{\mathrm{T}}$ is not affected by the noise mentioned above, which is bounded by $q/4$, and it does not affect the MSB. That is $\mathrm{MSB}(K) = \mathrm{MSB}(\boldsymbol{sy}^\top)$ is the proof of correctness.

*5.2. Security Analysis of Algorithm*

**Definition 3.** *If the advantage of any PPT adversary $\mathcal{A}$ in the above game is negligible, then the LWE-CP-ABE encryption scheme based on the LSSS access control structure is selectively secure under the linear independent restriction.*

**Definition 4.** *If the LWE assumption holds, the proposed LWE-CP-ABE scheme for all access structures is selectively secure.*

To prove Definition 3, the hybrid games start with the adversary sending an access policy to the challenger and the challenger sending back the public parameters to the adversary. Then, $\mathcal{A}$ requests to the challenger a polynomial number of secret keys. For each key query, the attacker sends a series of attributes $U \in \mathbb{U}$ that do not satisfy the access control policy $(\boldsymbol{M}, \rho)$. In addition, the row of the access control in the matrix $\boldsymbol{M}$ is marked by attribute in $U$; that is, the index of $\boldsymbol{M}$ in $\rho^{-1}(U)$ must be linearly independent. After that, the challenger replies to the corresponding user attribute secret key SK $\leftarrow$ **KeyGen**(MSK,$U$). Finally, $\mathcal{A}$ outputs its guess for the bit $b$ encrypted within the challenge ciphertext.

The advantage of the adversary $\mathcal{A}$ in this game is defined as

$$\mathrm{Adv}_{\mathcal{A}}^{\mathrm{LWE-CPABE,SEL-LI-CPA}}(\lambda) \triangleq |\mathrm{Pr}[b = b'] - 1/2| \tag{24}$$

Thus, how to generate the public parameters, secret keys, and challenge ciphertext in each hybrid game are described below.

**Hyb$_0$:** *This hybrid game corresponds to the true weak security selective game of the ABE scheme.*

| *Setup phase* | 5. $\forall u \in U : k_u = \hat{k}_u + \widetilde{k}_u$ |
|---|---|
| 1. $y \leftarrow \mathbb{Z}_q^n$. | 6. $\text{SK} = \left(\{k_u\}_{u \in \mathbb{U}}, t\right)$ |
| 2. $\{(A_u, T_{A_u})\}_{u \in \mathbb{U}} \leftarrow \textbf{EnTrapGen}(1^n, 1^m, q)$. | *Challenge phase* |
| 3. $\{H_u\}_{u \in \mathbb{U}} \leftarrow \mathbb{Z}_q^{n \times m}$. | 1. $s \leftarrow \mathbb{Z}_q^n$ |
| 4. $\text{PK} = \begin{pmatrix} n, m, q, \mathcal{X}_{lwe}, \mathcal{X}_1, \mathcal{X}_2, \mathcal{X}_{big}, \\ y, \{A_u\}_{u \in \mathbb{U}}, \{H_u\}_{u \in \mathbb{U}} \end{pmatrix}$. | 2. $\{v_j\}_{j \in \{2, \cdots, s_{\max}\}} \leftarrow \mathbb{Z}_q^m$ |
| *Key query phase* | 3. $\{e_i\}_{i \in [\ell]} \leftarrow \mathcal{X}_{lwe}^m$ |
| 1. $\{\hat{k}_u\}_{u \in \mathbb{U}} \leftarrow \mathcal{X}_{big}^m$. | 4. $\{e_i\}_{i \in [\ell]} \leftarrow \mathcal{X}_{big}^m$ |
| 2. $\hat{t} \leftarrow \mathcal{X}_1$. | 5. $\forall i \in [\ell] : c_i = s A_{\rho(i)} + e_i$ |
| 3. $t = (1, \hat{t})$. | 6. $\forall i \in [\ell] : \hat{c}_i = M_{i,1}\left(s y^\top, \overbrace{0, \cdots, 0}^{m-1}\right) + \left[\sum\limits_{j \in \{2, \cdots, s_{\max}\}} M_{i,j} v_j\right] - s H_{\rho(i)} + \hat{e}_i$ |
| 4. $\forall u \in U : \hat{k}_u \leftarrow$ $\textbf{EnSamplePre}\left(A_u, T_{A_u}, \sigma, t H_u^\top - \hat{k}_u A_u^\top\right)$ | 7. $\text{CT} = \left(\{c_i\}_{i \in [\ell]}, \{\hat{c}_i\}_{i \in [\ell]}, \text{MSB}\left(s y^\top\right) \oplus b\right)$ |

**Hyb$_1$:** *This game is similar to* **Hyb$_0$**. *The changes between* **Hyb$_0$** *and* **Hyb$_1$** *are merely syntactic and indistinguishable. The main difference is described as follows:*

1. In the **Setup Phase**, the generation of an additional matrix $\{B\}_{j \in \{2, \cdots, s_{\max}\}} \leftarrow \mathbb{Z}_q^{n \times m}$ is added between Steps 2 and 3;

2. In the **Challenge Phase**, let the original $\{v_j\}$ to $\{\hat{v}_j\}_{j \in \{2, \cdots, s_{\max}\}} \leftarrow \mathbb{Z}_q^n$. The vectors $\{\hat{c}_i\}$ are generated below using matrices while preparing the challenge ciphertext.

$$\forall i \in [\ell] : \hat{c}_i = M_{i,1}\left(s y^\top, \overbrace{0, \cdots, 0}^{m-1}\right) +$$
$$\left[\sum_{j \in \{2, \cdots, s_{\max}\}} M_{i,j} \hat{v}_j B_j\right] - s H_{\rho(i)} + \hat{e}_i$$

**Hyb$_2$:** *This game is the same as* **Hyb$_1$**, *except for the change of the generation of the matrix* $\{H_u\}$ *in the* **Setup Phase**.

$$1.\{H'_u\}_{u \in \rho([\ell])} \leftarrow \mathbb{Z}_q^{n \times m}$$
$$2.\forall u \in \rho([\ell]) : H_u = M_{\rho^{-1}(u),1}\left[y^\top \overbrace{\left|0^\top\right| \cdots \left|0^\top\right|}^{m-1}\right] + \sum_{j \in \{2, \cdots, s_{\max}\}} M_{\rho^{-1}(u),j} B_j + H'_u$$

The change between **Hyb$_2$** and **Hyb$_1$** is also only in syntax, so the two mixed games are indistinguishable.

**Hyb$_3$:** *This game is identical to* **Hyb$_2$**, *except for the change of the generation of the matrix* $\{H'_u\}$ *in the* **Setup phase***:*

$$1.\{R_u\}_{u \in \rho([\ell])} \leftarrow \{-1, 1\}^{m \times m}$$
$$2. H'_u = A_u R_u$$

According to Lemma 1, **Hyb$_3$** and **Hyb$_2$** are selectively indistinguishable.

**Hyb₄:** *This game is the same as **Hyb₃**, except for the change of the matrix in the **Setup Phase**:*

$$B' = [B_2'^\top | \cdots | B_{s_{\max}}'^\top, T_{B'}] \leftarrow$$

1. $\textbf{EnTrapGen}\left(1^{n(s_{\max}-1)}, 1^{m-1}, q\right):$
   $$\left\{B_j'\right\}_{j \in \{2,\cdots,s_{\max}\}} \in \mathbb{Z}_q^{n \times (m-1)}$$
2. $\left\{b_j'\right\}_{j \in \{2,\cdots,s_{\max}\}} \leftarrow \mathbb{Z}_q^n$
3. $\forall j \in \{2,\cdots,s_{\max}\} : B_j = [b_j'^\top | B_j']$

The indistinguishableness between **Hyb₄** and **Hyb₃** originates from the enhanced trapdoors lattice sampler function **EnLT** = (**EnTrapGen**, **EnSamplePre**).

**Hyb₅:** *This game is analogous to **Hyb₄**, except for the change of vector $\{k_u\}_{u \in U \cap \rho([\ell])}$ when answering the key query of the adversary $\mathcal{A}$:*

1. 
$$\widetilde{k}_u \leftarrow \textbf{EnSamplePre}$$
$$\left( \begin{array}{c} A_u, T_{A_u}, \sigma, \\ \left[ t\left( M_{\rho^{-1}(u),1}\left[ y^\top \Big| \overbrace{0^\top | \cdots | 0^\top}^{m-1} \Big] \right) \right]^\top + \\ \sum_{j \in \{2,\cdots,s_{\max}\}} t\left( M_{\rho^{-1}(u),j} B_j \right)^\top - \hat{k}_u A_u^\top \end{array} \right).$$

2. $k_u = \hat{k}_u + \widetilde{k}_u + t R_u^\top$

The indistinguishableness between **Hyb₅** and **Hyb₄** follows from Lemma 1.

**Hyb₆:** *This game is the same as **Hyb₅** except for the generation of vector $\hat{t}$ while answering the key query of the opponent $\mathcal{A}$. Let $d = (d_1, \cdots, d_{s_{\max}}) \in \mathbb{Z}_q^{s_{\max}}$ be a vector such that $d_1 = 1$ and for all $u \in U$, there is $\sum_{j \in [s_{\max}]} M_{\rho^{-1}(u),j} d_j = 0$. It is worth noting that due to the game restrictions, the set of the index of $M$ row in $\rho^{-1}(U)$ must be unauthorized to the access control policy $(M, \rho)$, thus ensuring the existence of the vector $d$.*

1. In the Key query phase, let $\hat{t} \leftarrow \mathcal{X}_1$ change to $\left\{f_j\right\}_{j \in \{2,\cdots,s_{\max}\}} \leftarrow \mathbb{Z}_q^n$

2. The vectors is $\hat{t} \leftarrow \textbf{EnSamplePre}\left( \begin{array}{c} B', T_{B'}, \sigma, \\ \left( \begin{array}{c} d_2 y + f_2 - b_2', \cdots, \\ d_{s_{\max}} y + f_{s_{\max}} - b_{s_{\max}}' \end{array} \right) \end{array} \right)$

The indistinguishableness between **Hyb₆** and **Hyb₅** originates from the enhanced trapdoors function **EnLT** = (**EnTrapGen**, **EnSamplePre**).

**Hyb₇:** *This game is the same as **Hyb₆**, except for the change of key module while answering the key query of the adversary $\mathcal{A}$:*

1. $\{z_u\}_{u \in U \cap \rho([\ell])} \leftarrow \mathbb{Z}_q^n$

2. $\left\{f_j\right\}_{j \in \{2,\cdots,s_{\max}\}} \leftarrow \mathbb{Z}_q^n$ exploits the constraint $\begin{array}{c} \forall u \in U \cap \rho([\ell]): \\ \sum_{j \in \{2,\cdots,s_{\max}\}} M_{\rho^{-1}(u),j} f_j = z_u + \hat{k}_u A_u^\top \end{array}$

3. $\begin{array}{c} \forall u \in U \cap \rho([\ell]): \\ \widetilde{k}_u \leftarrow \textbf{EnSamplePre}(A_u, T_{A_u}, \rho, z_u) \end{array}$

The change between **Hyb₇** and **Hyb₆** is only in syntax, so the hybrid games are indistinguishable.

**Hyb$_8$**: *This game is analogous to **Hyb$_7$**, except for the generation of vector $\{k_u\}_{u \in U \cap \rho([\ell])}$ while answering the key query of the adversary $\mathcal{A}$. The changes are described as follows:*

$$1. \left\{\widetilde{k}_u\right\}_{u \in U \cap \rho([\ell])} \leftarrow \mathcal{X}_2$$
$$\forall u \in U \cap \rho([\ell]):$$
$$2. \sum_{j \in \{2, \cdots, s_{\max}\}} M_{\rho^{-1}(u),j} \mathbf{f}_i = \widetilde{k}_u \mathbf{A}_u^\top + \hat{k}_u \mathbf{A}_u^\top$$

The indistinguishableness between **Hyb$_8$** and **Hyb$_7$** originates from the trapdoors function **EnLT = (EnTrapGen, EnSamplePre)**.

**Hyb$_9$**: *This game is similar to **Hyb$_8$**, except for the matrix generation $\{A_u\}_{u \in U \cap \rho([\ell])} \leftarrow \mathbb{Z}_q^{n \times m}$ during the Setup phase. The indistinguishableness between **Hyb$_9$** and **Hyb$_8$** originates from the enhanced trapdoors function **EnLT = (EnTrapGen, EnSamplePre)**.*

**Hyb$_{10}$**: *This game is the same as **Hyb$_9$** except for the generation of the vector $\{\hat{c}_i\}$ in challenging ciphertext during the Challenge phase, i.e.,*

$$1. \{e'_i\}_{i \in [\ell]} \leftarrow \mathcal{X}_{big}^m$$
$$2. \forall i \in [\ell] : \hat{e}_i = -e_i R_{\rho(i)} + e'_i$$

According to Lemma 1, **Hyb$_{10}$** and **Hyb$_9$** are selectively indistinguishable.

**Hyb$_{11}$**: *This game is the same as **Hyb$_{10}$**, except for the change in challenging ciphertext during the stage of Challenge:*

In the **Challenge Phase**, we have:

$$1. \tau \leftarrow \mathbb{Z}_q$$
$$2. \{\hat{v}'_j\}_{j \in \{2, \cdots, s_{\max}\}} \leftarrow \mathbb{Z}_q^n$$
$$3. \{e'_i\}_{i \in [\ell]} \leftarrow \mathcal{X}_{big}^m$$
$$4. \{c_i\}_{i \in [\ell]} \leftarrow \mathbb{Z}_q^m$$
$$\forall i \in [\ell]:$$
$$5. \hat{c}_i = [\sum_{j \in \{2, \cdots, s_{\max}\}} M_{i,j} \hat{v}'_j B_j] - c_i R_{\rho(i)} + e'_i$$
$$6. \text{CT} = \left(\{c_i\}_{i \in [\ell]}, \{\hat{c}_i\}_{i \in [\ell]}, \text{MSB}(\tau)\right)$$

According to the hypothesis of the LWE difficulty problem, **Hyb$_{10}$** and **Hyb$_9$** are selectively indistinguishable.

For any adversary $\mathcal{A}$ and any $x \in \{0, \cdots, 11\}$, let $p_{\mathcal{A},x}: \mathbb{N} \to [0,1]$ be a function. For all $\lambda \in \mathbb{N}$, we define the probability of the adversary winning the mixed game as $p_{\mathcal{A},x}(\lambda)$.

According to the definition of Hyb$_0$, for all $\lambda \in \mathbb{N}: |p_{\mathcal{A},0}(\lambda) - 1/2| = \text{Adv}_{\mathcal{A}}^{\text{LWE}-\text{CPABE,SEL}-\text{CPA}}(\lambda)$. In addition, for all $\lambda \in \mathbb{N}$, there is $p_{\mathcal{A},11} = 1/2$, there is no challenger's information about selecting the challenge bit in the challenge ciphertext in Hyb$_{11}$. Therefore, for all $\lambda \in \mathbb{N}$, there is: $\text{Adv}_{\mathcal{A}}^{\text{LWE}-\text{CPABE}}(\lambda) \leq \sum_{x \in [11]} |p_{\mathcal{A},x-1}(\lambda) - p_{\mathcal{A},x}(\lambda)|$.

In Hyb$_0$ and Hyb$_1$, the vector $\{\hat{v}_j\}_{j \in \{2, \cdots, s_{\max}\}}$ is uniformly and independently distributed on $\mathbb{Z}_q^n$, and the matrix $\{B_j\}_{j \in \{2, \cdots, s_{\max}\}}$ is uniformly and independently distributed on $\mathbb{Z}_q^{m \times n}$, so $\{\hat{v}_j B_j\}_{j \in \{2, \cdots, s_{\max}\}}$ is also uniformly and independently distributed on $\mathbb{Z}_q^m$. It is concluded that for any opponent $\mathcal{A}$, there is $p_{\mathcal{A},0}(\lambda) = p_{\mathcal{A},1}(\lambda)$.

According to the analysis, in Hyb1 and Hyb2, $p_{\mathcal{A},1}(\lambda) = p_{\mathcal{A},2}(\lambda)$.

**EnLT = (EnTrapGen, EnSamplePre)** satisfies the leftover hash theorem, so there is a negligible function $negl_i(\cdot)$ for any opponent $\mathcal{A}$ in Hyb$_3$ to Hyb$_{10}$. For all $\lambda \in \mathbb{N}$, there is $|p_{\mathcal{A},i-1}(\lambda) - p_{\mathcal{A},i}(\lambda)| \leq negl_i(\cdot)$.

Because the hypothesis of the LWE-hardness problem is true, there is a negligible function $negl_{11}(\cdot)$ for any PPT opponent $\mathcal{A}$, which satisfies $|p_{\mathcal{A},10}(\lambda) - p_{\mathcal{A},11}(\lambda)| \leq negl_{11}(\cdot)$ Therefore, the advantage of the opponent $\mathcal{A}$ in the mixed game is 0.

### 5.3. Analysis of Security on the Blockchain

This section will introduce the common blockchain attack models and how this scheme can resist these typical attacks.

Conspiracy attacks: In the distributed key management system, the attacker collusion attack would require M nodes to collect the shares of the master key. This means the attacker must pay a high cost for collecting all the pieces together. Furthermore, each user attribute secret key will generate a uniformly and independently distributed random vector $\hat{t} \leftarrow \mathcal{X}_1$ when generating in our scheme, and the random vector of any user secret key is different; therefore, the user attribute secret key is hidden by information theory from anyone, so that the conspiracy attack cannot be realized.

Middleman attacks: middleman attack means that the attacker sets up independent exercises at both ends of the correspondence, exchanges the received data, and monitors or tampers the information. In this scheme, all the correspondences among the nodes are conducted in the form of transactions. The transaction is signed by the initiator using the blockchain secret key, and the returned secret data is encrypted by the other party's public key, so the middleman cannot pass the verification by tampering with the address or forging the signature. Therefore, this scheme can greatly prevent middleman attacks.

Link attacks: link attacks mean the adversary searches for the user's private data by linking multiple transactions with the same address. In our scheme, the related authorization process of user attributes can be determined by the user, and the user can choose to disclose his identity attributes or hide the identity information to protect privacy. In addition, the relevant data is the ciphertext on the blockchain, and the security is guaranteed by the algorithm security, so the attacker cannot get the relevant information of the user, thus resisting the link attacks.

### 5.4. Comparative Analysis of Performance

In this section, a comparative study of the proposed approach with other classical CP-ABE schemes is presented in Table 1. It is observed that all the works employ LSSS-based access structures, with [44] utilizing a hierarchical access structure. FAN [7] uses q-PBDHE as its hardness problem without adopting multi-authority mechanisms, which means that the secret key cannot be cracked and the one-way function cannot be inverted within polynomial time. However, there is a negligible probability of cracking it. Sammy [9] utilizes d-DDH hardness problems as security assumptions, providing an ECC scheme instead of a bilinear pairing in terms of computational complexity overhead. Additionally, it employs a multi-authority CP-ABE scheme with a hierarchical LSSS access structure. Datta [35] and Mohammad [45], on the other hand, separately employ LWE and R-LWE CP-ABE to enhance their security through a multi-authority scheme, achieving resistance against quantum attacks, although they are not based on blockchain and introduce a significant increase in computational complexity.

**Table 1.** Comparison with related schemes.

| Scheme | [7] | [9] | [35,44] | Ours |
|---|---|---|---|---|
| Blockchain | √ | √ | × | √ |
| Privacy Preservation | √ | √ | √ | √ |
| Anti-Quantum Attack | × | × | √ | √ |
| Provable Security | Not always provable | Provably secure | Provably secure | Provably secure |
| Hardness Problem | q-PBDHE | d-DDH problem | LWE, R-LWE | LWE |
| Multi-Authority | × | √ | √ | √ |
| Access Structure | LSSS | LSSS | LSSS, Hierarchical LSSS | LSSS |

Note: √ means the solution satisfies this characteristic, × means it does not.

Our scheme extends the previous works by utilizing a lattice-based CP-ABE scheme in conjunction with blockchain-based PRE threshold networks to manage the master secret key. It provides automated verification and consensus of transactions through smart contracts,

ensuring confidentiality, accountability, and traceability of transactions. Moreover, it is a security type based on the worst-case problem, where the cryptographic algorithm needs to be solved in the worst-case scenario if it is cracked. Lastly, other CP-ABE schemes described in the references require more expensive operations such as multiplication, modulus exponentiation, and bilinear mapping, while our scheme only requires addition operations with lower expense, thereby significantly improving operational efficiency.

### 5.5. Analysis of Experimental Simulation

To conduct a thorough evaluation of the proposed scheme's practical effectiveness, a series of experiments were performed in this paper. All experiments were conducted on a computer running the Win10 operating system with the following hardware configuration: Intel Core i7-8750H CPU, 2.20 GHz clock frequency, and 8.0 GB RAM. The paper utilized the PBC [46] library based on pairings and the PALISADE API. A simulation framework was built using the C++ programming language. Additionally, a blockchain virtual network based on Ganache [47] was set up, and the Remix platform provided by Ethereum was used for compiling and deploying smart contract code online. When selecting the performance index, the scheme in this paper does not change the transaction process of the blockchain network but only encrypts the data in the network by LWE-CP-ABE, which will not affect the operation efficiency of the blockchain network. Therefore, this paper only evaluates the performance index of the LWE-CP-ABE scheme.

In this experiment, the plaintext data is set to 308B. The scheme in this paper does not need complex operations such as modulus exponentiation and bilinear mapping. Still, only the addition operation, so the time cost in each stage is better than that of the scheme in reference [7].

As shown in Figures 4 and 5, with the continuous increase of attributes in the system, the time spent by the data owner in the setup and encryption under the blockchain in this scheme is shorter than that in the scheme of reference [7], which greatly improves the overall efficiency of the system.

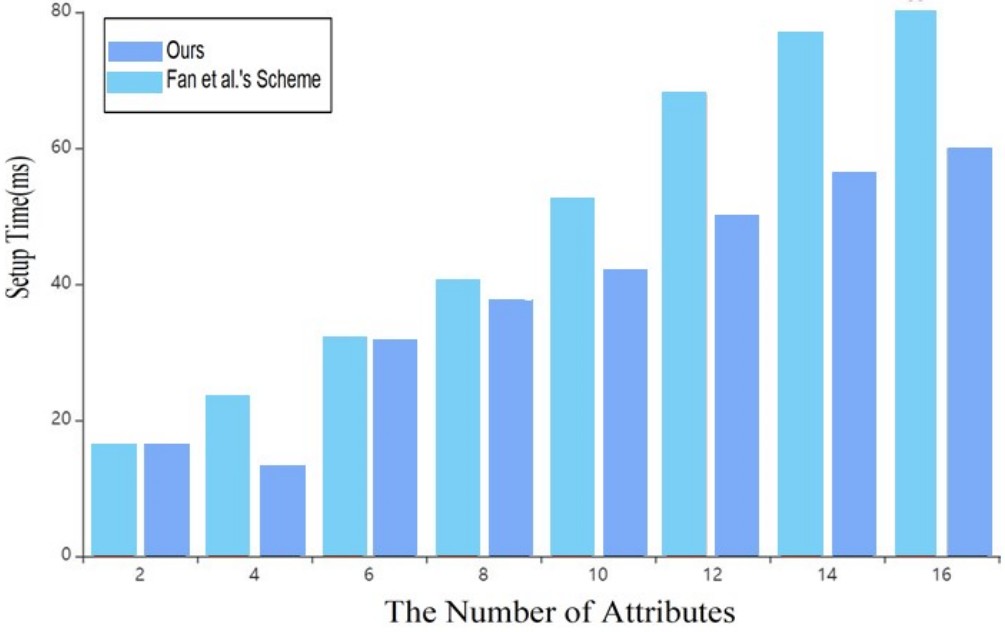

**Figure 4.** Time cost of Setup [8].

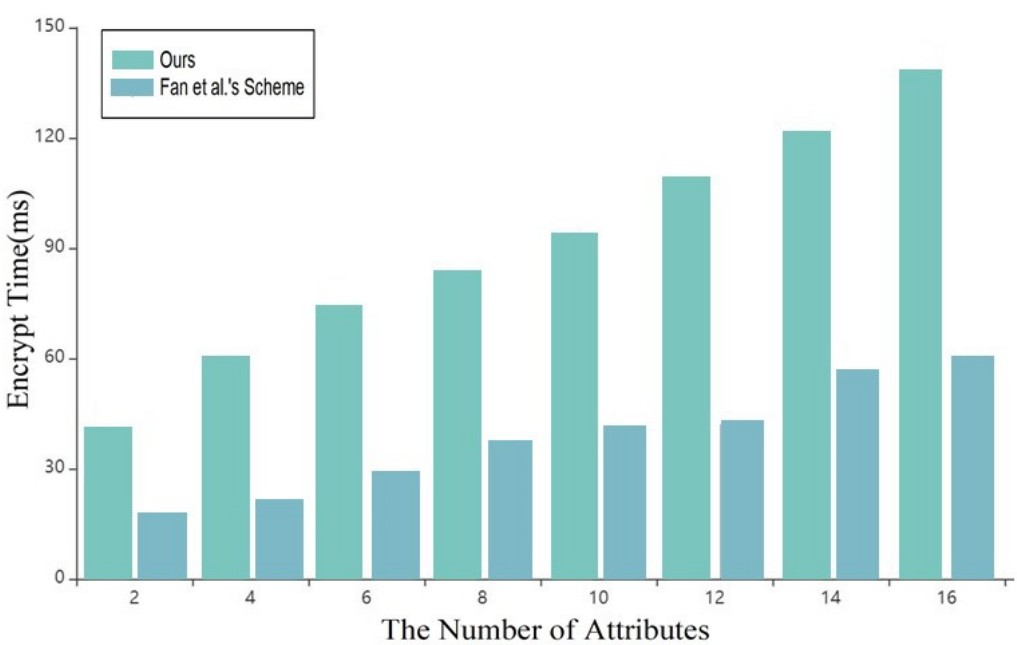

**Figure 5.** Time cost of Encryption [8].

As shown in Figure 6, in the key generation stage, with the continuous increase of attributes, this scheme is superior to that in reference [7], and it will be more practical in key generation with multi-user and multi-attribute sets.

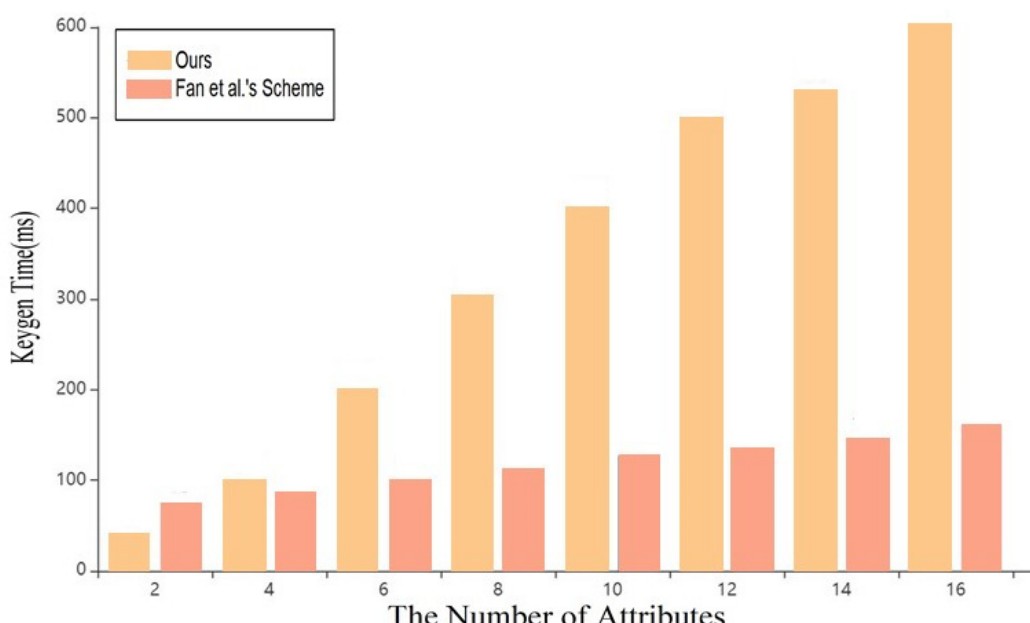

**Figure 6.** Time cost of key generation [8].

As shown in Figure 7, in the decryption stage, the scheme in this paper takes about 0~2 ms, which is far lower than the scheme in reference [7]. Therefore, it is more suitable for blockchain data sharing.

In addition, considering the pre-quantum blockchain system, the transaction cost for executing the smart contract related to signature verification on KMS is around 1,091,035 gas, with an execution cost of 965,801 gas. The transaction cost for executing the smart contract related to the correctness of the re-encrypted results is 3,597,371 gas, with an execution cost of 3,300,467 gas. That means that we can create a multi-lateral market to provide incentive mechanisms to enhance the security of access control based on the blockchain system.

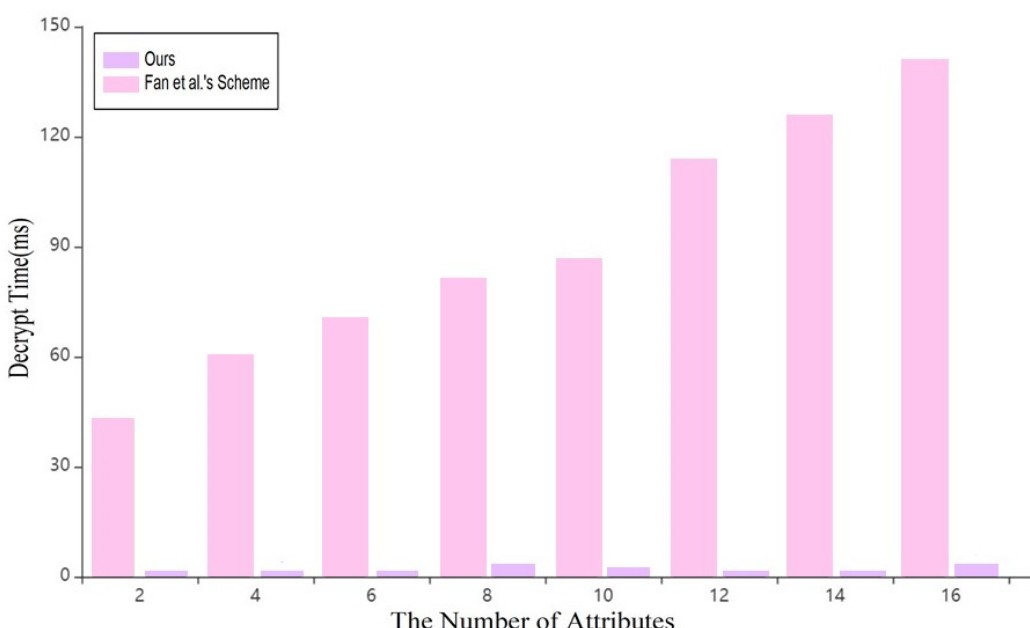

**Figure 7.** Time cost of Decryption [8].

## 6. Conclusions

Privacy protection technology on the blockchain has always been a significant factor in data access and sharing. With the rapid development of quantum computation, the traditional public key cryptosystem based on number theory cannot resist quantum attacks. Therefore, this paper effectively integrates blockchain technology with the lattice attribute-based encryption algorithm and proposes a data-sharing scheme based on LWE-CP-ABE using blockchain technology. This paper improves the CPABE scheme put forward by Datta, designs the ABE algorithm against quantum attacks with the renewable strategy, and fulfills the dynamic protection of data. The lattice-based fine-grained access control of CP-ABE is constructed through the decentralized KMS and formatted transaction structure over a pre-quantum cryptographic blockchain system, which not only ensures the characteristics of the post-quantum CP-ABE algorithm against quantum attacks but also provides the traceable on-chain transactions of the participant's activities. The simulation experiment shows that the performances (as can be seen in Figures 4–7) is superior to the traditional CP-ABE scheme. Meanwhile, decentralized KMS powered by blockchain technology enables the distribution of the key management process between Data Owners and Data Users by applying a threshold proxy re-encryption scheme to eliminate any risk of centralization and collusion.

As a future research direction, we plan to extend our protocol to achieve IND-CCA security in the post-quantum setting. Specifically, we aim to develop a decentralized multi-authority CP-ABE scheme based on blockchain resistant to chosen-ciphertext attacks and can support any non-monotone access structure. Furthermore, we aim to explore comprehensive countermeasures against combined attacks, such as Differential Power Analysis (DPA) and Differential Fault Analysis (DFA). We will investigate and implement countermeasures based on techniques such as Time Insertion (TI) and error detection schemes to enhance the security of our system. These efforts will contribute to strengthening the resilience of our protocol against various differential analysis attacks, ensuring the reliability and security of the system.

**Author Contributions:** Conceptualization, T.C.; methodology, Z.R.; software, Y.Y.; formal analysis, J.Z. (Jie Zhu); project administration, J.Z. (Jinyi Zhao). All authors have read and agreed to the published version of the manuscript.

**Funding:** This research was funded by the National Natural Science Foundation of China, grant number No. 61961042 and No.71964037; Yunnan Key Laboratory of Blockchain Application Technology grant number No. 202105AG070005 and No. YNB202108; Yunnan International Joint Research and Development Center for Cross-border Trade and Financial Blockchain, grant number No. 202203AP140010; Kunming International (Foreign-oriented) Science and Technology Research and Development Center for Blockchain Technology in South Asia and Southeast Asia, grant number No. GHJD-2022006; Research on Key Technologies of Cross-Border Trade Blockchain for RCEP, grant number No. 202202AD080011; Scientific Research Foundation of Yunnan Education Department, grant numbers No. 2023Y0657 and No. 2023Y0675.

**Institutional Review Board Statement:** Not applicable.

**Informed Consent Statement:** Not applicable.

**Data Availability Statement:** The data used to support the findings of this study are available from the corresponding author upon request.

**Conflicts of Interest:** The authors declare no conflict of interest.

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
