# Peer review of "Lattices-Inspired CP-ABE from LWE Scheme for Data Access and Sharing Based on Blockchain"

_applsci, doi:10.3390/app13137765_

Round 1

Reviewer 1 Report

The article is very interesting, and the authors show that they know blockchain technologies very well.
The references are interesting (37).
It might be interesting for the authors to expand on the following points:
- Improve the explanation of the "Notations" section to better understand your proposal
- Improve the explanation of functions 16 and 21 (page 9 and page 13) (unclear).
- Improve the explanation of the "Comparative analysis of performance" section (unclear).
- Have you tested your method in a real application (and how?)
- On page 5, you say "consists mainly of six procedures the following syntax" but there are 7 (check this part).

Good

Author Response

Dear Reviewer,

   Thank you for your feedback and suggestions regarding our article. We appreciate your positive comments about the paper and the authors' understanding of blockchain technologies. We have carefully addressed each of the points you raised and made the necessary improvements, which are highlighted in red below:

  1. We have improved the explanation of the "Notations" section to provide a clearer understanding of our proposal for readers.
  2. The descriptions of functions 16 and 21 (page 9 and page 13) have been revised to provide more clarity and ensure better comprehension of these functions.
  3. The explanation of the "Comparative analysis of performance" section has been enhanced to clarify the differences between the listed approaches and highlight the advantages of our proposed method.
  4. To evaluate and analyze the practical effectiveness of our proposed method, we conducted a series of experiments. All experiments were performed on computers running the Windows 10 operating system with an Intel Core i7-8750H CPU, 2.20GHz, and 8.0GB RAM. We utilized the paired-based PBC cryptographic library and PALISADE API, implementing a simulation experimental framework using the C++ programming language. Additionally, we set up a blockchain virtual network based on Ganache and used the Remix online smart contract compilation and deployment platform provided by Ethereum to compile and deploy smart contract code.
  5. We have carefully reviewed the relevant section and can confirm that there are indeed only six procedures: Setup, KeyGen, Enc, Dec, AccGen, and AccUpdate. The statement on page 5 indicating "consists mainly of six procedures the following syntax" is accurate.

    Thank you once again for your valuable feedback. We believe that these revisions have significantly improved the clarity and comprehensibility of our paper. If you have any further suggestions or questions, please feel free to let us know.

Best regards,

Zhixin Ren

Reviewer 2 Report

Why the research in the paper is important:

To address the quantum attacks on number theory-based cipher-text policy attribute based encryption(CP-ABE), and to avoid private key leakage problem relying on a trustworthy central authority, authors propose a lattice-inspired CP-ABE scheme for data access and sharing based on blockchain in this paper. Firstly, a CP-ABE based algorithms using Learning With Errors (LWE) assumption are constructed, which is selective security under linear independence restriction in the random oracle model. Secondly, the blockchain nodes can act as a distributed key manage server to offer control over master keys used to generate private keys for different data users that reflect their attributes through launching transactions on blockchain system. Finally, we develop smart contracts for proving correctness of proxy re-encryption(PRE) and provide auditability for the whole process of data sharing. Compared with the traditional CP-ABE algorithm, the post-quantum CP-ABE algorithm can significantly improve the computation speed according to the result of the functional and experimental analysis. Moreover, the proposed blockchain-based CP-ABE scheme not only provides multi-cryptography collaboration to enhance the security of data access and sharing, but also reduce average transaction response time and throughput. ---------------------

The topic is very important, NIST announced the winners in 2022, this is the hottest topic in crypto now.

See my comments regarding PQC and SCA and lightweight crypto detailed below.

- With any new security measure implementation, you need to make sure you provide benchmark for active/passive side-channel attacks (SCAs). Fault attack and power analysis attack and countermeasures need to be mentioned. Mention and add a paper about "fault detection of ring-LWE on FPGA" too.

- With the advent of post-quantum cryptography (PQC), it is better to add some relevant works to make sure you cover that topic too. This is the hottest topic in cryptography now. When PQC replaces ECC/RSA every security application from smart phones to block chains will be affected. With PQC, add a paper on these topics separately: (a) Curve448 and Ed448 on Cortex-M4, (b) SIKE on Cortex-M4, (c) SIKE Round 3 on ARM Cortex-M4, (d) Kyber on 64-Bit ARM Cortex-A, (e) Cryptographic accelerators on Ed25519.

- NIST lightweight standardization was finalized in Feb. 2023. Also mention fault attacks as side-channel attacks, these topics to explore and add a paper references on each of these separately: (a) Fault detection of architectures of Pomaranch cipher, (b) reliable architectures of grostl hash, (c) fault diagnosis of low-energy Midori cipher, (d) fault diagnosis of RECTANGLE cipher.

- References are not uniformly formatted.

- Please add a subsection and one or more future works for enhancing your presentation

- DPA+DFA for example can be mounted at the same time and their combined countermeasures (for example TI and Error Detection Schemes) can be used for thwarting attacks in combined manner (need to be discussed). Adding a paragraph suffices.

Author Response

Dear Reviewer,

       Thank you very much for your feedback on our paper. We greatly appreciate your valuable comments, and we have made targeted revisions accordingly. All modifications have been highlighted in yellow for easy identification.

     We acknowledge the importance of the issues you raised regarding post-quantum cryptography (PQC), side-channel attacks (SCA), and lightweight cryptography. PQC is currently one of the hottest topics in the field of cryptography, and its adoption will have a significant impact on the security applications ranging from smartphones to blockchain. We have included several relevant papers related to PQC, covering topics such as (a) Curve448 and Ed448 on Cortex-M4, (b) SIKE on Cortex-M4, (c) SIKE Round 3 on ARM Cortex-M4, (d) Kyber on 64-Bit ARM Cortex-A, and (e) Cryptographic accelerators on Ed25519. Additionally, we have mentioned fault attacks as a form of side-channel attacks and added references for each topic, including (a) fault detection of architectures of Pomaranch cipher, (b) reliable architectures of grostl hash, (c) fault diagnosis of low-energy Midori cipher, and (d) fault diagnosis of RECTANGLE cipher. These relevant references have been cited in the paper with the corresponding numbers [36-43].

    Furthermore, we have made uniform formatting adjustments to the references. Additionally, we have revised the section on future works and expanded it to include countermeasures against combined DPA and DFA attacks. We aim to enhance the security of our proposed scheme by exploring the design of joint countermeasures, such as Trusted Initialization (TI) and Error Detection Schemes, to thwart attacks in such scenarios.

       Thank you once again for your guidance and suggestions. Your feedback has been instrumental in improving our paper.

Sincerely,
Zhixin Ren

Round 2

Reviewer 2 Report

Comments are addressed.